# Crude Heparin Preparations Unveil the Presence of Structurally Diverse Oversulfated Contaminants

**DOI:** 10.3390/molecules24162988

**Published:** 2019-08-17

**Authors:** Aline Mendes, Maria C. Z. Meneghetti, Marcelly Valle Palladino, Giselle Zenker Justo, Guilherme L. Sassaki, Jawed Fareed, Marcelo A. Lima, Helena B. Nader

**Affiliations:** 1Departamento de Bioquímica, Instituto de Farmacologia e Biologia Molecular, Escola Paulista de Medicina, Universidade Federal de São Paulo, Rua Três de Maio, 100 – São Paulo, SP 04044-020, Brazil; 2Departamento de Ciências Farmacêuticas, Campus Diadema, Universidade Federal de São Paulo, Rua Três de Maio, 100 – São Paulo, SP 04044-020, Brazil; 3Departamento de Bioquímica, Setor de Ciências Biológicas, Centro Politécnico, Universidade Federal do Paraná, Avenida Coronel Francisco H. dos Santos, 100, Caixa Postal 19031, Curitiba, PR 81531-980, Brazil; 4Department of Molecular Pharmacology & Therapeutics, Hemostasis & Thrombosis Research Laboratories, Loyola University Medical Center, 2160 S. First Avenue Maywood, IL 60153, USA; 5Molecular & Structural Biosciences, School of Life Sciences, Keele University, Huxley Building, Keele, Staffordshire ST5 5BG, UK

**Keywords:** oversulfated chondroitin sulfate, heparin and oversulfated chondroitin sulfate anticoagulant activities, ^1^H NMR and COSY, FACE, heparinases

## Abstract

Nowadays, pharmaceutical heparin is purified from porcine and bovine intestinal mucosa. In the past decade there has been an ongoing concern about the safety of heparin, since in 2008, adverse effects associated with the presence of an oversulfated chondroitin sulfate (OSCS) were observed in preparations of pharmaceutical porcine heparin, which led to the death of patients, causing a global public health crisis. However, it has not been clarified whether OSCS has been added to the purified heparin preparation, or whether it has already been introduced during the production of the raw heparin. Using a combination of different analytical methods, we investigate both crude and final heparin products and we are able to demonstrate that the sulfated contaminants are intentionally introduced in the initial steps of heparin preparation. Furthermore, the results show that the oversulfated compounds are not structurally homogeneous. In addition, we show that these contaminants are able to bind to cells in using well known heparin binding sites. Together, the data highlights the importance of heparin quality control even at the initial stages of its production.

## 1. Introduction

Heparin is a linear, highly sulfated glycosaminoglycan (GAG) chain of various lengths with molecular weights varying from 2000 to 40,000 Da [1,2] and is composed of a repeating disaccharide units of 1,4 linked α-l-iduronic or β-d-glucuronic acid (d-GlcA), and α-d-glucosamine (d-GlcN). The predominant substitution pattern comprises 2-*O*-sulfation of the iduronate residues and *N*- and 6-*O*-sulfation of the glucosamine residues [3].

Heparin is an established anticoagulant drug for the prevention and control of thrombotic events owing to its interaction with a number of proteins of the blood clotting cascade, notably antithrombin increasing its inhibitory effect on thrombin and other coagulation serine proteases [4]. Antithrombin is the main regulator of coagulation proteases in vertebrates, acting upon thrombin and factors XIIa, XIa, IXa and Xa activities. The inhibitory effect is due to the fact that heparin induces conformational modifications in antithrombin, favoring its interaction with the serine proteases forming a ternary complex [5,6]. Heparin is also capable of potentiating cofactor II activity, which is another serpin of the coagulation cascade, specific for thrombin inhibition. In addition, upon heparin treatment, endothelial cells enhance the synthesis and release of an antithrombotic heparan sulfate as well as tissue factor pathway inhibitor (TFPI) [7].

Pharmaceutical heparin is largely obtained from bovine and porcine intestinal mucosa by a multi-step extraction process that involves proteolysis, anion exchange chromatography or quaternary ammonium complexes, ethanol precipitation (where raw heparin is collected), bleaching, fractionation by alcohol precipitation and drying [8,9]. Considering heparin as a natural pharmaceutical of animal origin, the amount of potential impurities is relatively high when compared to a fully synthetic drug. Such impurities may be in the form of other GAGs, such as chondroitin, dermatan and heparan sulfates, residual protein or nucleic acids, solvent and salts. Since this process is quite complex, it requires carefully executed procedures and effective quality control monitoring to avoid the co-purification of impurities and contaminating species [10,11,12].

In the last decade there has been concern about the safety of heparin due to events related to heparin contamination. In 2008, significant numbers of adverse clinical responses such as arrhythmia, nausea, edema, hypotension, anaphylactic reactions associated with contaminated heparin were reported, which led to deaths in the United States [13,14,15,16]. 

The contaminant was characterized as an oversulfated chondroitin sulfate (OSCS). The structural unit of the OSCS corresponds to a 2,3-disulfated glucuronic acid linked to a 4,6-disulfated N-acetylgalactosamine. The glycosidic bonds are β-1,4 between galactosamine and glucuronic acid and β-1,3- between glucuronic acid and galactosamine. OSCS presents structural and physicochemical similarities with heparin, which makes these molecules easily co-purified [13,17]. 

From the identification of this component, several studies have shown the biological effects of this contaminant proposing analytical methods to evaluate the purity of the produced heparin [12,16,18,19]. However, it has not been clarified if OSCS has been added to the purified heparin preparation, or it has been introduced during the production of the raw heparin. Thus, in the present study commercial raw heparin preparations from the Techpool Company (Guangdong, China) and final product from Changzhou SPL Company (Jiangsu, China), extracted and purified from porcine intestinal mucosa were analyzed. The presence and influence of the contaminants on heparin quality control methods as well as some pharmacological effects were investigated.

## 2. Results and Discussion

### 2.1. Identification of Contaminants in Crude Heparins and Final Product by Agarose Gel Electrophoresis

The degree of purity of the heparin preparations was initially evaluated by agarose gel electrophoresis in two different buffers (PDA and discontinuous Ba/PDA). Figure 1 shows the pattern of electrophoretic mobility for selected samples. When comparing the results observed for standard heparin and the samples, the presence of other components which bind to the quaternary amine precipitating in the gel and exhibit metachromatic activity with toluidine blue becomes evident. In the discontinuous Ba/PDA buffer, this is most evident due to a clear separation of slow migrating bands which are not present in standard heparin. The figure also shows that there are differences in the amounts of such contaminants among both the crude and final product heparin. 

The electrophoretic behavior of all crude heparin and final heparin batches in different buffers are depicted in Appendix A (Appendix A) and the percentage of contaminants in each sample obtained after densitometry of the PDA gels are shown in Appendix A. 

Among the 19 crude heparin batches analyzed, the contaminants were present in 17 samples and the relative amounts ranged from 7.6 to 28.1%. Curiously, the contaminant is still present in 8 out of 12 of final product heparin preparations where the percentage of contaminants ranged from 3.7 to 25.2%.

### 2.2. Scanning UV Spectroscopy Detects Contaminants in Heparin Preparations

Figure 2 shows the scanning UV spectroscopy from 190 to 320 nm of selected batches of crude and final product heparin. Standard heparin UV spectra shows a narrow peak of 190–210 nm whereas, on the other hand, contaminated heparins have a broader peak around 200–220 nm, as well as an additional broad signal around 240–260 nm (nucleic acids, peptides) [16]. This experiment was performed for all crude and final samples (data not shown).

### 2.3. Degradation with Lyases–Heparinase and Heparitinase II 

Figure 3 shows the degradation profile of both crude and final product heparin preparations using individually heparinase and heparitinase II. Heparinase action upon standard heparin leads to the formation of unsaturated products, especially tetrasaccharide and trisulfated disaccharide. On the other hand, the results show that in the presence of the contaminants there is a decrease in the degradation of heparin by heparinase, suggesting an inhibition of the enzyme. This is most evident for crude samples C8 and C9, as well as samples F8 and F9 of final heparin preparation. Apparently, the degradation of heparin by heparitinase II is not inhibited by the presence of the contaminant (Figure 3). These experiments were performed for all crude and final samples (data not shown).

### 2.4. Isolation of the Contaminants

The contaminants present in the crude samples (C8, C9 and C16) as well as final product (F4, F8 and F9) were isolated by ion exchange column chromatography in order to investigate their structure. The elution profile of the ion exchange column for the C8 sample is shown in Figure 4. It is clear the presence of peaks eluted with different salt concentrations. As depicted in Figure 4A, three peaks were selected for further analyses. The electrophoretic mobility in agarose gel shows that peak III has a lower migration, suggesting higher sulfation when compared to other GAGs, and was denominated OSC8 (Figure 4B). The same procedure was performed for other batches and the purified contaminants isolated for further analyses. 

### 2.5. Structural Characterization of Heparin Contaminants by Nuclear Magnetic Resonance (NMR)

The isolated contaminants purified from crude and final heparin batches were structurally characterized by nuclear magnetic resonance (^1^H NMR and COSY).

Figure 5 shows the ^1^H NMR spectra for standard OSCS. As already demonstrated by others [13,16], the signal in the 2.16 ppm region which corresponds to N-acetyl present in galactosamine residues is clear. The N-acetyl signal for glucosamine residues is depicted at 2.04 ppm being typical of heparin molecules, whereas for dermatan sulfate the N-acetyl also present in galactosamine residues gives a signal at 2.08 ppm (results not shown). Other OSCS characteristic signals are present in the region of 4.87 ppm, corresponding to the substituted glucuronic acid.

Sulfated contaminants isolated from some batches of crude and final heparin display similar ^1^H NMR spectra of standard OSCS (Figure 5). Nevertheless, contaminants purified from other batches (OSC8 and OSC9) in the ^1^H NMR spectra show signals in the region from 4.0 to 4.55 ppm, that may correspond to non-fully substituted glucuronic acid residues and are not detected in OSCS (Figure 6). 

The confirmation of the partial sulfation of the isolated compound was performed by two-dimensional ^1^H-^1^H NMR correlation spectroscopy (COSY). Figure 7 shows the COSY spectra for the standard OSCS as well as for the isolated OSC9. The U2-U3 correlation signal is shifted, as expected, downfield since position 3 is replaced by a sulfate (electronegative) group. On the other hand, in the COSY spectrum of the OSC9 isolated compound, we observed an upfield U2-U3 correlation signal which suggests the incomplete sulfation at the 3-position of some glucuronic acid residues.

### 2.6. Exhaustive Degradation of OSC8 with Chondroitinases AC and ABC

The structural characteristics of the OSC8 was further investigated by degradation with chondroitinases AC and ABC. Under conditions of exhaustive degradation (72 h and 0.3 enzyme units) no product formation was observed by SAX-HPLC ion exchange column chromatography, even though the polysaccharide is non-sulfated at position C-3 of the glucuronic acid (Appendix A). On the other hand, the enzymes acted upon chondroitin sulfate (CS) and dermatan sulfate (DS), yielding the expected products. 

The degradation products were further analyzed by FACE (Figure 8). The products formed were derivatized with AMAC and subjected to polyacrylamide gel electrophoresis and visualized by UV light absorption. Again, it becomes clear that the OSC8 is not a substrate for the enzymes, since no products could be detected even by this methodology that enhances the detection of products (fluorescence x UV absorbance). On the other hand, the typical products were obtained by the action chondroitinases AC and ABC (Figure 8). According to Calabro and coworkers [20], AMAC gives the same molar fluorescence value for every derivatized saccharide independent of its chemistry and is 100 times more sensitive than the detection by UV absorbance of the unsaturation.

### 2.7. Effect of the Isolated Contaminants on Degradation of Heparin by Heparinase 

As previously demonstrated, batches with contaminants show decrease in the degradation of heparin by heparinase (Figure 3). Thus, a kinetic study using purified OSCSs from both crude and final product preparations was performed to investigate this putative inhibitory effect upon the degradation of heparin by heparinase (Figure 9). The results showed that indeed the sulfated compounds isolated from crude heparins (OSC8 and OSC9) and final product (OSF4 and OSF8) as well as standard OSCS inhibit the action of heparinase upon heparin in a dose and time dependent manner. All compounds at the lowest tested concentration (0.1 µg/mL) inhibited heparinase activity from 66% to 85%. 

### 2.8. APTT

APTT assay shows that both crude heparin samples contaminated with OSCS (C8, C9, C16) or the final heparin product also contaminated with OSCS (F4, F8, F9) display lower anticoagulant activity when compared to pure heparin sample (Appendix A, Appendix A).

### 2.9. Effects of the Contaminants on Endothelial Cells

The effect of crude and final product heparins and respective purified sulfated contaminant were investigated using endothelial cells (EC) in culture. The results regarding cell viability show no effect of the compounds in all tested concentrations (Appendix A, Appendix A). 

It is known that heparin binds to endothelial cells. Thus, in order to test if the isolated contaminants were capable of binding to EC, competitive experiments were performed. Biotinylated heparin binds to EC in a dose dependent manner and such binding is decreased in the presence of excess molar amounts of all the tested contaminants as also observed for standard heparin (Figure 10). 

### 2.10. Discussion

This paper was designed to clarify whether the OSCS found in heparin pharmaceutical preparations was added to the purified heparin preparation or whether it was introduced during the production of raw heparin. Several methodologies were used to evaluate the degree of purity of these different preparations, as well as the structural characteristics and some pharmacological activities of the preparations and isolated contaminants. 

Initially, the identification of sulfated contaminants in crude heparins and final products was demonstrated using agarose gel electrophoresis in two different buffers (Figure 1). The electrophoresis in PDA showed contaminants with lower migration than standard heparin, suggesting higher interaction with the diamine of the buffer, indicating different size and or distribution of charges when compared to heparin. The presence of the contaminants became even more evident in the discontinuous barium acetate/PDA system, where standard heparin is fractionated into three components (slow, intermediate and fast) according to their mobility, due to differences in size and charge. The contaminants displayed lower migration than the slow-moving component of heparin, again indicating the presence of compounds with high degree of sulfation. Interesting, since the contaminants were already found in the crude heparin preparations, we can conclude that these compounds were co-purified with heparin, and intentionally added to the raw preparation, since these types of polymers do not occur naturally. 

The typical UV scanning spectra of GAGs show a peak around 190 nm resulting from the electronic transition at the carboxyl of the iduronic and glucuronic acids present in the polymers as well as the acetoamide of the hexosamines (Figure 2, standard porcine and bovine heparins). The spectra obtained for both crude and final product heparin samples showed a broader and less defined peak around 200–220 nm corresponding to the carboxyl groups of uronic acids of the structurally modified chondroitin sulfates, as delineated for standard oversulfated chondroitin sulfate. In addition, these samples show an additional peak with absorption in the region of 260 nm related to the UV spectrum of nucleic acids and peptides. This method of analysis allows a rapid assessment of the presence of contaminating sulfated polymers in commercial preparations of heparin since the broader peak may indicate the presence of two components. This set of results again shows definitively that the contamination of heparin by oversulfated compounds occurs during the production of crude heparin.

The structure of these heparins was investigated by incubation with heparinase and heparitinase II. The results show the formation of unsaturated products, mainly tetrasaccharides and trisulfated disaccharide by the action of heparinase and disulfated and N-sulfated disaccharides after heparitinase II degradation (Figure 3). However, the amounts of products formed upon the action of heparinase was dependent on the relative load of the contaminant present in each sample, that is, higher amounts of contaminant, less formation of unsaturated products, which suggests that the contaminant is inhibiting the heparinase action (Figure 3). 

Furthermore, this hypothesis is confirmed by kinetic studies where standard heparin was incubated with heparinase in the presence of the purified contaminants (Figure 9). These results indicate that these contaminants can bind to heparinase due to their structural characteristics interfering with enzyme activity. In previous work we have shown that the inhibition of heparinase by OSCS is irreversible, since addition of heparin does not dislodge the inhibitor [21]. Similar results were also found by other authors [22]. 

On the other hand, our data shows that the contaminant present in raw heparin preparations and heparin final product do not affect heparitinase II action upon heparin. The differences observed in the figure are related to the fact that bovine heparin is a better substrate for heparitinase II compared to porcine heparin. The results from other authors, performing kinetics studies of heparitinase II in the presence of OSCS indicate that the enzyme is inhibited [22]. These apparent discrepancies may be related to the types of experiments performed by the groups. 

The structure of the purified sulfated contaminants was characterized by NMR. The results showed the presence of typical oversulfated chondroitin sulfate, that is a compound containing 2,3-disulfated β-d-glucuronic acid and 4,6-disulfated β-d-*N*-acetylgalactosamine (Figure 5). The data also showed the presence of other contaminants with different structure where glucuronic acid residues are present bearing lower sulfation pattern (^1^H NMR signal at 4.55 ppm) (Figure 6). Confirmation of the partial sulfation of the isolated compound was performed by two-dimensional NMR by COSY (Figure 7). The COSY spectra for OSCSs shows that U2-U3 correlation signal is shifted, as expected, downfield since position 3 is replaced by a (electronegative) sulfate group culminating in the deshielding of such a proton in (Figure 7). On the other hand, the COSY spectra for OSC9 (Figure 7) and for OSC8 (data not shown) showed an upfield U2-U3 correlation signals, possibly related to non-sulfated glucuronic at the 3-position. Thus, for the OSCS we only have the presence of 2,3-*O*-disulfated glucuronic acid residues, whereas for the OSC9 compounds we observed residues of 2-*O*-sulphated glucuronic acid (Figure 7). This structural difference may be related to a partial sulfation of the polymer.

The NMR data led us to investigate whether these sulfated contaminants would be substrate for chondroitinases AC and ABC under conditions of exhaustive degradation. For the sulfated contaminants, no unsaturated products were detected at 232 nm after SAX-HPLC ion-exchange chromatography, contrasting with the data obtained for standard chondroitin sulfate and dermatan sulfate (Appendix A). This result was further confirmed by a more sensitive method based on the derivatization with a fluorophore followed by FACE analysis of the enzymatic products. Again, the contaminant samples of OSCS with a lower sulfation pattern, where glucuronic acid residues are not persulfated, were not processed by both enzymes, whereas the products for the standard galactosaminoglycans are clearly depicted (Figure 8). 

Data from the literature show that chondroitin sulfate E isolated from squid presents in some regions of the polymer, sulfation pattern similar to OSCS, and is partially degraded by chondroitinase AC. A chondroitin sulfate isolated from *Litopenaeus vanamei* shrimp has been described in the literature, with disaccharide units consisting of disulfated *N*-acetylgalactosamine 4,6 linked to a 2,3-disulfated glucuronic acid in some regions of the polysaccharide chain [23]. This CS is also resistant to the action of chondroitinase under exhaustive degradation conditions. The results suggest that sulfation in glucuronic acid residues renders the substrate resistant to the action of enzymes [24]. 

As already observed by other works, OSCS decreases heparin anticoagulant activity by in vitro assays that measure the intrinsic pathway, such as the activated partial thromboplastin time [25,26]. Since OSCS, as well as the OSCS with a lower degree of sulfation, mimic the structure of heparin and display similar effects on some anticlot assays [27], we decided to investigate if these compounds could interact with the heparin binding sites in endothelial cells [28]. The results showed that indeed both compounds bind to endothelial cells and are dislodged by heparin, thus indicating their binding to the same cellular sites. Heparin when injected to human beings will interact endothelial cells to produce antithrombotic effects. Thus, the fact that the contaminants interact with endothelial cells and dislodge heparin from its binding site is of clinical interest for further understanding the side effects of these contaminants. 

The data clearly showed that heparin adulteration occurred at the initial steps of heparin production, suggesting manipulation to alter anticoagulant activity. Due to the complex chemical nature of heparin and the OSCS contaminants, the purity of the heparin sample needs to be performed by a set of methods to guarantee quality control [16,19,29].

## 3. Materials and Methods

### 3.1. Heparin Samples and Standard OSCS

Commercial preparations of raw heparin (crude heparin) were obtained from the Techpool Bio-pharma Company (Guangdong, China) and purified heparin (final product) from Changzhou SPL Company (Changzhou City, Jiangsu, China) (Appendix A). All samples were manufactured in 2008 and received for analysis in our laboratory in 2011. There is no correspondence between the batches of raw heparin and final product, but they were all obtained at the same period of time from the same companies. Standard heparin from porcine mucosa (183 IU/mg) and bovine mucosa (165 IU/mg) was obtained from Kin Master Produtos Químicos Ltda. (Passo Fundo, RS, Brazil). Semi-synthetic standard OSCS was prepared as previously described [30].

### 3.2. Agarose Gel Electrophoresis

Sulfated glycosaminoglycans (GAGs) were initially identified by agarose gel electrophoresis using 0.05 M 1,3-diaminopropane acetate (PDA) buffer system, pH 9.0 (Dietrich & Dietrich, 1976) or discontinuous buffer 0.04 M barium acetate, pH 5.8/0.05 M 1,3-diaminopropane acetate, pH 9.0 (Ba/PDA) [31]. Aliquots of the compounds (1.0 mg/mL; 5–10 µL) were applied to a 0.55% agarose gel and ran for 1 h at 100 V in the cold. The GAGs were precipitated in the gel with 0.1% cetyltrimethylammonium bromide solution. After 2 h, the gels were dried under heat and air, stained for 15 min with 0.1% toluidine blue in 1% acetic acid in 50% ethanol, and destained with the same solution without the dye. GAGs were quantified by densitometry at 525 nm were using Quick Scan 2000 equipment, Helena Laboratories (Beaumont, TX, USA).

### 3.3. Scanning UV Spectroscocy

Samples of heparins and other GAGs (1mg/mL in water) were scanned from 190 to 320 nm with 1 nm resolution at 120 nm/min using a Perkin-Elmer Lambda 25 UV/VIS spectrometer (Waltham, MA, USA) [16].

### 3.4. Ion Exchange Chromatography for Isolation of the Sulfated Contaminants

Isolation of the sulfated contaminant was achieved by Q-Sepharose ion exchange chromatography using HPLC chromatography (AKTA Purifier) eluted with a 2M NaCl gradient, from 0 to 85% using 5 mL/min flow rate for both crude heparins and final products. The samples were collected in 5 mL fractions, dialyzed against distilled water and lyophilized for further analysis.

### 3.5. Degradation with Different Glycosaminoglycan Lyases 

Samples of crude and purified heparins (100 µg) were incubated with 3 mIU of heparinase (renamed heparinase I or heparin lyase I) or heparitinase II (renamed heparinase II or heparin lyase II) from *Flavobacterium heparinum* in 0.1M ethylenediamineacetate (EDA) buffer pH 7.0 at 30 °C for approximately 18 h [32]. Samples of the isolated contaminant (OSCS) (100 µg) were incubated with 30 mU of the chondroitinase AC (from *Arthrobacter aurescens*) and ABC (from *Proteus vulgaris*) in 50 mM sodium acetate buffer pH 8.0 at 37 °C for approximately 24 h. In some experiments, the incubation was performed for 72 h with addition of the enzymes every 24 h. Analytical separation of the products formed by the action of the specific enzymes was performed both by paper and HPLC chromatography as previously described [33,34]. The products were separated on descending paper chromatography (Whatman nr. 1) using isobutyric acid: 1.25 M (5:3, *v*/*v*) NH_4_OH as solvent for 18 h. The chromatogram was oven dried with heating (80 °C) and ventilation. The products were detected by UV absorption (232 nm), revealed with silver nitrate in alkaline medium and fixed with 5% sodium thiosulfate. For HPLC, the products were resolved in Zorbax SAX column (4.6 × 150 mm; 5 µm particle), with 0–1.5 M NaCl gradient with a flow of 1 mL/min for 30 min and the unsaturated products monitored by absorbance at 232 nm.

### 3.6. Fluorophore Assisted Carbohydrate Electrophoresis (FACE)

The analysis of disaccharides derived from glycosaminoglycans was also performed by FACE as described [20,35]. The degradation with chondroitinases yields unsaturated products containing free reducing group, which were derivatized with AMAC (2-aminoacridone). The saccharide-AMAC derivatives were separated by polyacrylamide gel electrophoresis. The derivatization with AMAC consists of the reaction between saccharides with free reducing aldehyde group and a primary amino group of a fluorescent compound with production of a Schiff base. This reaction is reversible and needs to be accompanied by reduction to the secondary amine derivative using sodium cyanoborohydride. For the reaction, aliquots containing 10 to 200 nmol saccharides with free reducing moieties were dried under vacuum and derivatized with 5 µL of 50 mM AMAC solution (250 nmol) in DMSO: acetic acid (85:15). After 15 min at room temperature 5 µL of freshly prepared 1M sodium cyanoborohydride solution was added. This derivatization mixture was maintained for 16 h at 37 °C and afterwards 30 µL of 30% glycerol added. Aliquots of the derivatized samples (2 µL) were then analyzed by electrophoresis on 20% polyacrylamide gel in Tris-borate glycine buffer (0.6M/0.5M/0.5M) pH 8.3. The bands were visualized after exposure to UV lamp at 365 nm and photographed (MF-ChemiBism DNR Biosystems, Neve Yamin, Israel).

### 3.7. Inhibitory Effect of OSCS on Heparinase Activity

The inhibitory effect of OSCS on heparin degradation by heparinase was assayed in the presence of 1085 µM heparin, 10 mM Ca^2+^ in 50 mM HEPES, pH 7.0 at 30 °C. The assay was performed using an UV spectrophotometer at 232 nm with temperature regulation. Incubation was performed using both recombinant heparinase [21] and purified heparinase from *F*. *heparinum* [33] in 50 mM Hepes buffer, 150 mM NaCl, pH 7.0 in the presence of different concentrations of OSCS. The enzyme was preincubated with 10 mM calcium acetate at 30 °C, and then standard heparin was added. The formation of the unsaturated products was monitored at 232 nm for 10 min. Afterwards, different amounts of the isolated contaminants were added, and the formation of products monitored for another 10 min at 232 nm.

### 3.8. Nuclear Magnetic Resonance (NMR) for Structural Characterization of the Contaminants in Heparin Preparations

The structural characterization of the contaminating polymers isolated from the adulterated heparins was performed by NMR spectroscopy. Six individual sulfated compounds isolated from the crude heparins and final product were analyzed. ^1^H NMR spectroscopy and ^1^H^1^H two-dimensional COSY homology was performed according to the USP monograph for sodium heparin [36]. Solutions of 20 mg/mL of each sample in deuterium oxide (99.9%) were analyzed for the acquisition of free induction decay (FID) using 16 scans, 90° pulse and 20 s delay at 37 °C in a Bruker Spectrometer. All spectra were obtained with a Bruker 400 MHz, 600 MHz AVANCE II or III NMR spectrometer (Bruker GmbH, Silberstrei-fen, Germany). Special attention was given to the spectral region in which the N-acetyl group with chemical shift of ~2.00 ppm indicates the presence of other species of GAGs. Typical ^1^H NMR spectra of non-fractionated heparins show signals corresponding to the trisulfated disaccharide (α-l-iduronate 2-*O*-sulfate → 4α-d-glucosamine N, *O*-disulfated), which is the major repeating unit in heparin [16]. 

### 3.9. APTT Assay

Activated partial thromboplastin time was performed in a Dade Behring Coagulation Analyzer BFT II, Siemens Healthcare Diagnostics, Inc (Tarrytown, NY, USA), following the protocol of the APTT, Helena Laboratories (Beaumont, TX, USA). The assays were performed with platelet-poor plasma, prepared from the collection of plasma in the presence of sodium citrate (3.8%), centrifuged for 300 rpm, 15 min and packed in plastic tubes. For activated partial thromboplastin time (APTT), platelet-poor plasma (50 μL) was incubated with cephalin (50 μL) and different concentrations (0.625 μg/mL to 5.0 μg/mL) of crude heparin samples (C8, C9, C16), final heparin samples (F4, F8, F9), standard porcine heparin and standard oversulfated chondroitin (50 μL) at 37 °C for 2 min. Next, 10 mM calcium chloride was added to determine clot formation time. Analyzes were performed in triplicate. The anticoagulant activity is expressed as time in sec.

### 3.10. Endothelial Cells Culture 

Endothelial cells (EC) derived from rabbit aorta were maintained in F12 medium supplemented with 10% fetal bovine serum (FBS) and penicillin (50U/L)/streptomycin (50 µg/L) [37]. For the subculture, cells near confluence were detached with a solution of pancreatin (2.5%) diluted 1:10 (*v*/*v*) in EBSS, suspended in F12 medium as described above and aliquots of 3 × 10^5^ cells transferred to a new plate (35 mm). Cells were maintained at 37 °C under moist atmosphere containing 2.5% CO_2_. The medium was replaced after 24 h and each subculture performed near the confluence. 

### 3.11. Cell Viability Assay 

Cytotoxic effects of the compounds were evaluated using EC, which were plated in 96-well microplates (1 × 10^4^ cells per well in F12 medium supplemented with 10% FBS and penicillin/streptomycin, enriched F12) and maintained for 24 h in an incubator at 37 °C in 2.5% CO_2_ atmosphere. The cells were then exposed to different concentrations of the various batches of heparins or isolated contaminants in 150 µL of enriched F12 for 24 h at 37 °C in 2.5% CO_2_ atmosphere. Afterwards the medium was aspirated and Alamar Blue^®^ (resazurin sodium) prepared in serum-free culture medium (0.1 mg/mL; 200 µL/well) added. The cells were then incubated for 4 h at 37 °C in an atmosphere of 2.5% CO_2_. Next, 100 µL of the medium were transferred to 96-well microplates and the reduction of the rezasurin salt in resorufin measured by fluorescence analysis (λExc = 560 nm; λEm = 590 nm) using FlexStation^3^ PlateReader (Molecular Devices, Sunnyvale, CA, USA). Viability represents the mean and standard error of two independent experiments carried out in sextuplicate.

### 3.12. Competitive Binding Assay between Heparin and OSCSs

The binding assays were performed as previously described [28]. Briefly, EC (1 × 10^4^ per well) were cultured for 3 days in 96 well plates. Afterwards, the medium was removed, and the cells washed with PBS containing 1% BSA at 4 °C. The cells were then exposed for 1 h, at 4 °C to different concentrations of biotinylated heparin in the presence of different amounts of the isolated contaminants and 100 times molar excess of standard heparin After, the cells were washed with PBS containing 1% BSA. The binding of the biotinylated heparin to the cells was detected by incubation with Europium conjugated streptavidin (1:5,000 in PBS) for 40 min at 4 °C and the fluorescence analyzed on fluorescence reader Victor^2^ (Wallac, Turku, Finland).

## Figures and Tables

**Figure 1 molecules-24-02988-f001:**
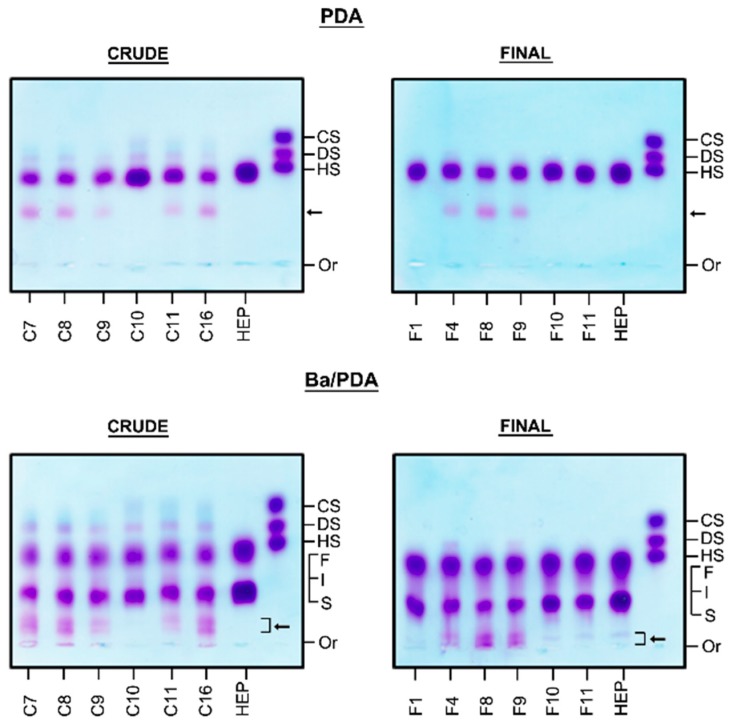
Identification of contaminants in selected samples of crude and final heparins by agarose gel electrophoresis. Aliquots (5 µL, 5–10 µg) of heparin samples were applied to 0.55% agarose gel and submitted to electrophoresis in 0.05M PDA or Ba/PDA discontinuous system as described in Methods. The arrow shows the presence of slow migrating bands which are not present in standard heparin. In the discontinuous Ba/PDA buffer this is most evident due to a clear separation of slow migrating bands. The figure also shows differences in the amounts of these contaminants among both the crude and final product heparin. C7, C8, C9, C10, C11 and C16: crude heparin samples; F1, F4, F8, F9, F10 and F11: final product heparin samples; HEP: standard porcine heparin; CS/DS/HS: standard mixture of chondroitin sulfate (CS), dermatan sulfate (DS), and heparan sulfate (HS); S: heparin slow moving component, I: heparin intermediate moving component and F: heparin fast moving component; Or: origin.

**Figure 2 molecules-24-02988-f002:**
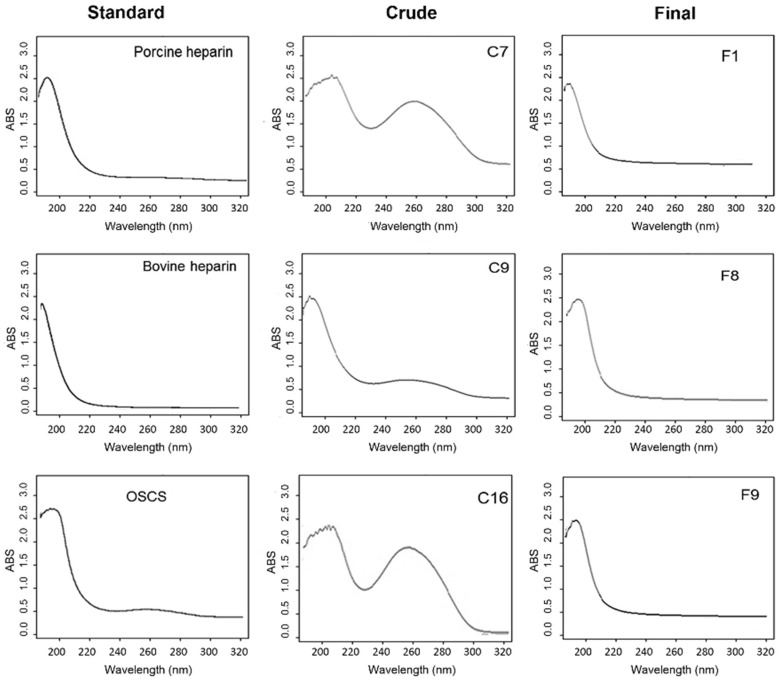
Analysis of crude and final heparins by UV scanning spectroscopy (190–320 nm). UV spectra were performed on UV/VIS spectrophotometer using 1 mg/mL solution in water in the range of 190–320 nm, with a reading rate of 120 nm/min with a resolution of 1 nm at room temperature. Standard heparin UV spectra shows a narrow peak of 190–210 nm whereas, on the other hand, contaminated heparins have a broader peak around 200–220 nm, as well as an additional broad signal around 240–260 nm (nucleic acids, peptides). C7, C9 and C16: crude heparin samples; F1, F8 and F9: final heparin samples).

**Figure 3 molecules-24-02988-f003:**
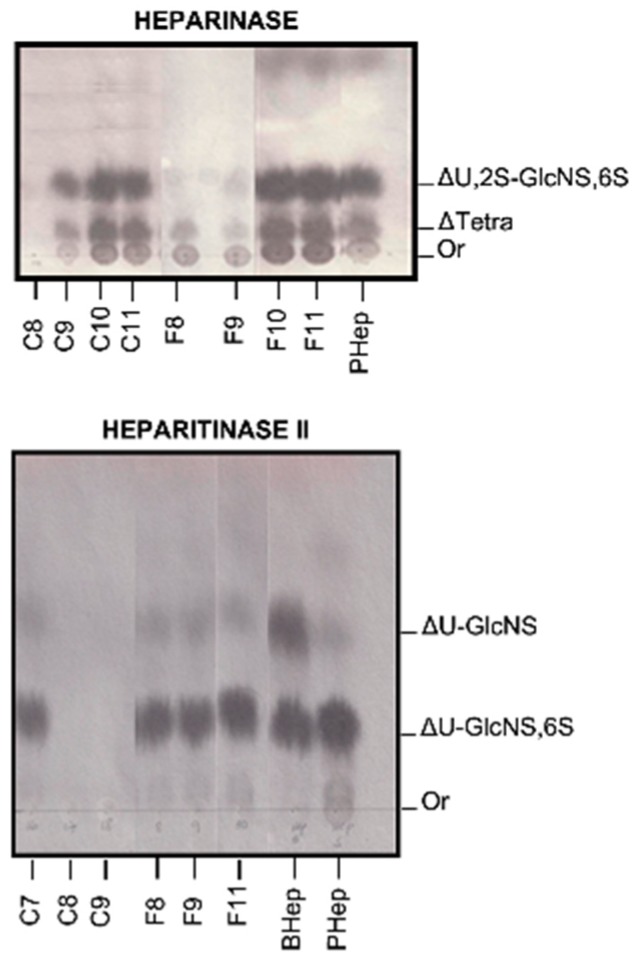
Analysis of heparinase and heparitinase II products from crude and final heparin samples. Crude and final heparin samples were incubated with heparinase and heparitinase II and the degradation products analyzed by paper chromatography as described. The results show that in the presence of the contaminants there is a decrease in the degradation of heparin by heparinase, suggesting an inhibition of the enzyme. BHep: bovine standard heparin; PHep: porcine standard heparin; Crude heparins (C7 to C11); Final heparin samples (F8, F9 and F11); ΔTetra: unsaturated pentasulfated tetrasaccharide; (ΔU2S-GlcNS,6S): unsaturated trisulfated disaccharide; (ΔUGlcNS,6S): disulfated unsaturated disaccharide; (ΔU-GlcNS): unsaturated N-sulfated disaccharide; Or: origin.

**Figure 4 molecules-24-02988-f004:**
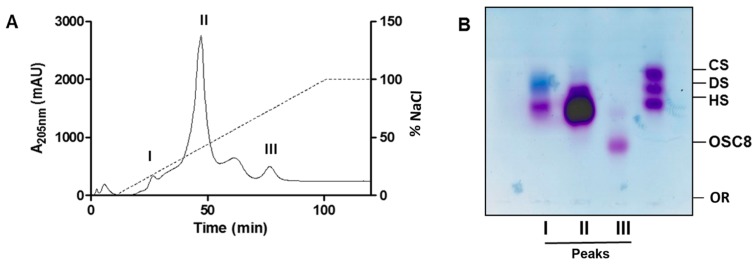
Purification of sulfated contaminants by ion exchange chromatography. (**A**) Elution profile of the crude heparin preparation C8 on Q-Sepharose ion exchange chromatography. I, II and III: the fractions shown in A were combined according to their elution profile and named peaks I to III. (**B**): Agarose gel electrophoresis in PDA buffer of Q-Sepharose fractions. Peak III corresponds to the contaminant OSC8, isolated from C8 crude heparin. CS/DS/HS: standard mixture of chondroitin sulfate (CS), dermatan sulfate (DS) and heparan sulfate (HS); Or: Origin.

**Figure 5 molecules-24-02988-f005:**
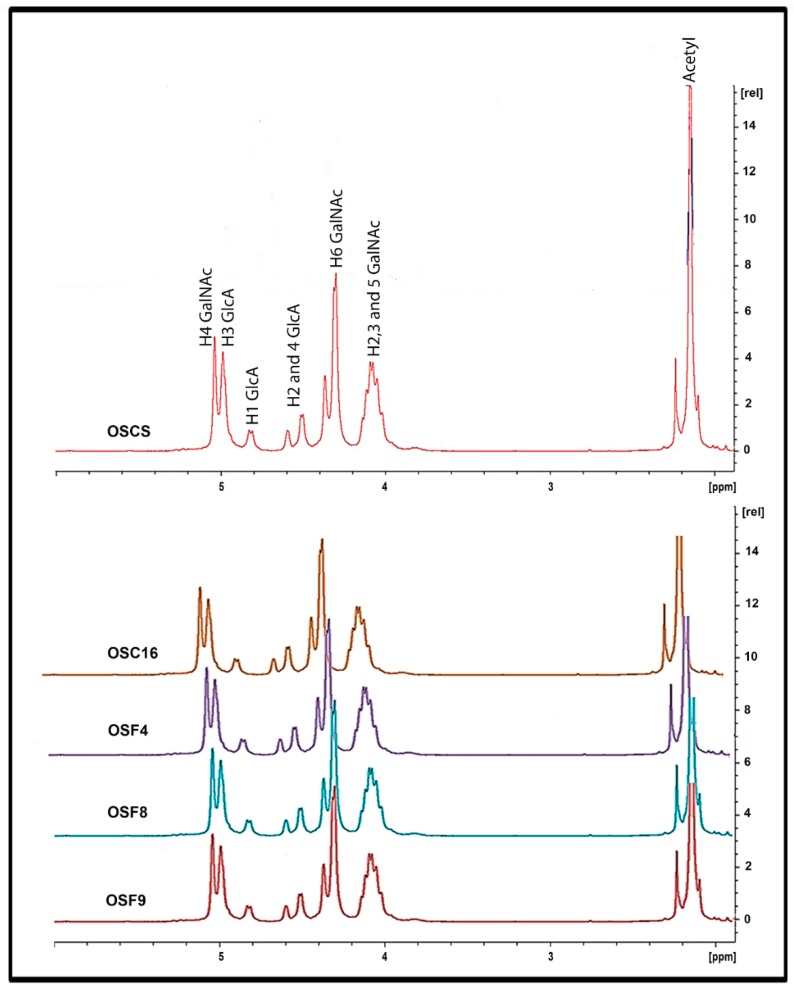
^1^H-NMR spectra of standard OSCS and sulfated contaminants isolated from crude (OSC16) and final heparin samples (OSF4, OSF8 and OSF9). The isolated contaminants purified from crude and final heparin batches were structurally characterized by nuclear magnetic resonance (^1^H NMR). Solutions of 20 mg/mL of each sample in deuterium oxide (99.9%) were analyzed for the acquisition of free induction decay (FID) using 16 scans, 90° pulse and 20 s delay at 37 °C in a Bruker Spectrometer DRX-500. Sulfated contaminants purified from batches of crude (OSC16) and final heparin (OSF4, OSF8 and OSF9) display similar ^1^H NMR spectra of standard OSCS.

**Figure 6 molecules-24-02988-f006:**
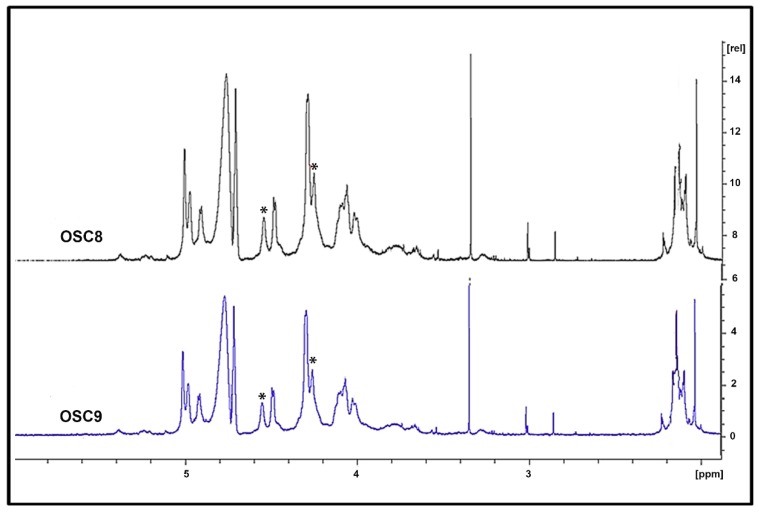
^1^H-NMR spectra of sulfated contaminants isolated from crude heparins OSC8 and OSC9 show distinct structure. The isolated contaminants purified were structurally characterized by nuclear magnetic resonance (^1^H NMR). Solutions of 20 mg/mL of each sample in deuterium oxide (99.9%) were analyzed for the acquisition of free induction decay (FID) using 16 scans, 90° pulse and 20 s delay at 37 °C in a Bruker Spectrometer DRX-500. Sulfated contaminants purified from crude batches (OSC8 and OSC9) in the ^1^H NMR spectra show signals in the 4.55 ppm region, marked with asterisks (*), that may correspond to non-fully substituted glucuronic acid residues and are not detected in OSCS.

**Figure 7 molecules-24-02988-f007:**
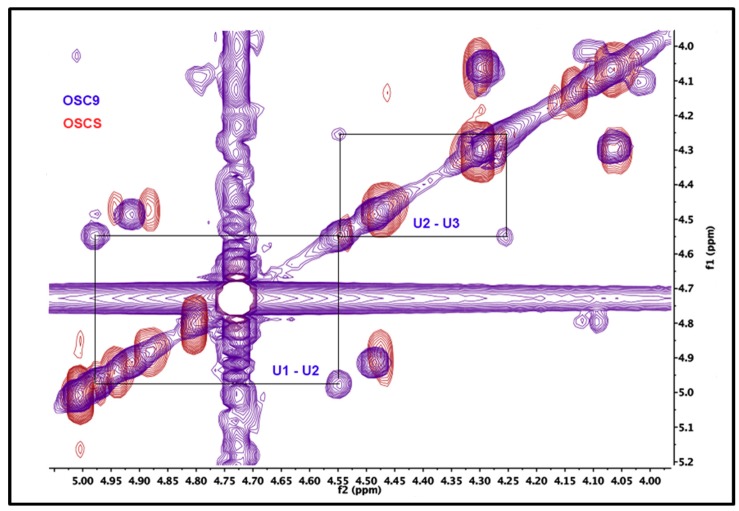
Two-dimensional NMR COSY spectra for the standard OSCS and the isolated OSC9. ^1^H NMR spectroscopy and ^1^H^1^H two-dimensional COSY homology was performed according to the USP monograph for sodium heparin. In the OSCS COSY spectra (red), the U2-U3 correlation signal is shifted, as expected, downfield since position 3 is replaced by a sulfate (electronegative) group. On the other hand, the OSC9 COSY spectra (blue) shows a downfield unlinked U2-U3 correlation signal which suggests the complete non-sulfation at the 3-position of the glucuronic acid residues.

**Figure 8 molecules-24-02988-f008:**
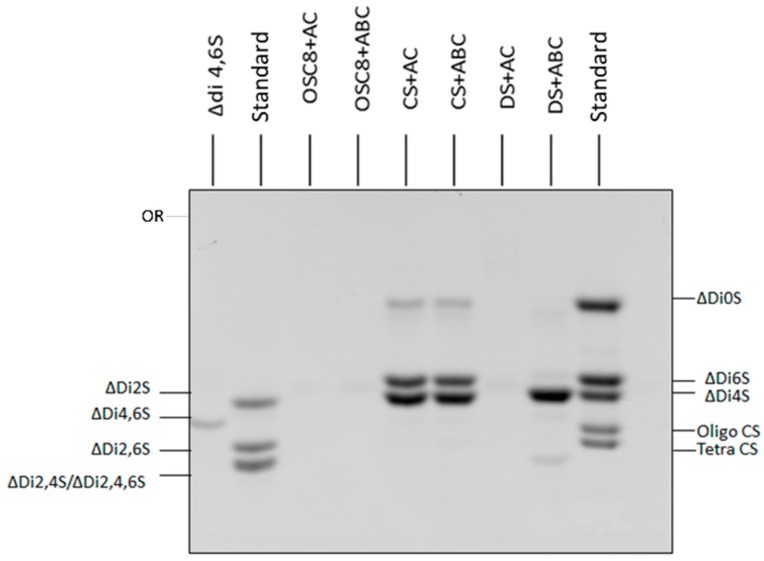
Analysis by Fluorophore Assisted Carbohydrate Electrophoresis (FACE) of the products formed from OSC8 by the exhaustive degradation with Chondroitinases AC and ABC. The structural characteristics of the OSC8 were investigated by degradation with chondroitinases AC and ABC under exhaustive degradation conditions (72 h and 0.3 enzyme units). The products formed were derivatized with AMAC and analyzed by polyacrylamide gel electrophoresis in Tris Borate pH 8.3 buffer, and displayed by UV light absorption, as described in Methods. OSC8 is not a substrate for the enzymes, since no products could be detected by this methodology that enhances the detection of products. ΔDi2S: unsaturated 2-sulfated disaccharide; ΔDi4S: unsaturated 4-sulfated disaccharide; ΔDi6S: unsaturated 6-sulfated disaccharide; ΔDi4,6S: unsaturated 4,6-disulfated disaccharide; ΔDi2,6S: unsaturated 2,6-disulfated disaccharide; ΔDi2,4S: unsaturated 2,4-disulfated disaccharide; ΔDi2,4,6S: unsaturated 2,4,6-trisulfated disaccharide; ΔDi0S: non-sulfated unsaturated disaccharide; Standard: mixture of standard unsaturated disaccharides; Tetra: tetrasaccharides; Oligo: oligosaccharides; OSC8: sulfated contaminant purified from C8 crude heparin; CS: standard chondroitin sulfate; DS: standard dermatan sulfate; AC: chondroitinase AC; ABC: chondroitinase ABC; Or: Origin.

**Figure 9 molecules-24-02988-f009:**
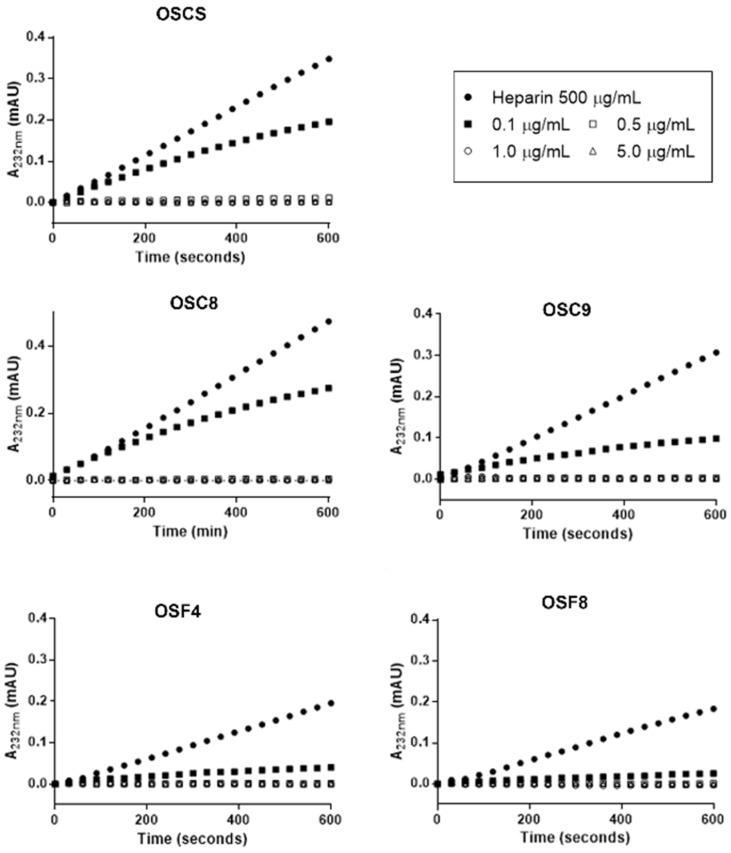
Kinetics of heparin degradation by heparinase in the presence of standard OSCS and sulfated contaminants isolated from crude and final heparin. The assay was performed using an UV spectrophotometer at 232 nm with temperature regulation. Incubation was performed using both recombinant heparinase and purified heparinase from *F*. *heparinum* in 50 mM Hepes buffer, 150 mM NaCl, pH 7.0. The enzyme was preincubated with 10 mM calcium acetate at 30 °C, and then standard heparin added. The formation of the unsaturated products was monitored at 232 nm for 10 min. Afterwards, different amounts of the isolated contaminants were added, and the formation of products monitored for 10 min at 232 nm. The sulfated contaminants isolated from crude heparins (OSC8 and OSC9) and final product (OSF4 and OSF8) as well as standard OSCS inhibit the action of heparinase upon heparin in a dose and time dependent manner. OSCS: standard oversulfated chondroitin sulfate; OSC8: sulfated contaminant purified from crude heparin C8; OSC9: sulfated contaminant purified from crude heparin C9; OSF4: sulfated contaminant purified from final heparin F4; OSF8: sulfated contaminant isolated from final heparin product F8.

**Figure 10 molecules-24-02988-f010:**
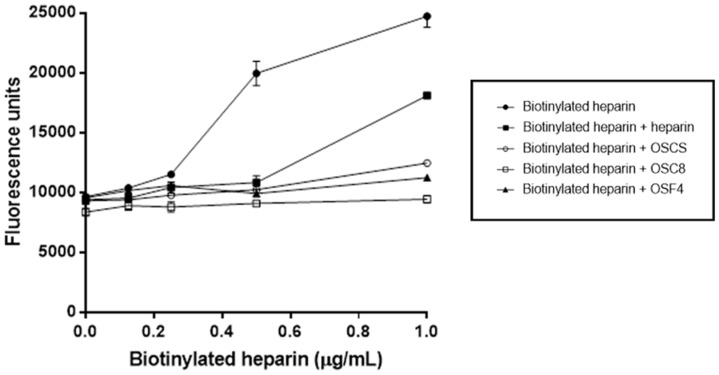
Isolated sulfated contaminants bind to endothelial cells. Endothelial cells were cultured in 96-well plates in F12 medium containing 10% FBS. The cells were exposed to different concentrations of biotinylated heparin plus molar excess of standard heparin or sulfated contaminants. Binding of biotinylated heparin to cells was detected by incubation with Europium-conjugated streptavidin and fluorescence analyzed on a fluorescence reader. Biotinylated heparin binds to EC in a dose dependent manner. This binding is decreased in the presence of excess molar amounts of all the tested contaminants as also observed for standard heparin. Heparin: standard heparin; OSCS: standard oversulfated chondroitin sulfate; OSC8: sulfated contaminant purified isolated from crude heparin preparation C8; OSF8: sulfated contaminant purified from final heparin product F8.

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
