# Peer review of "Crude Heparin Preparations Unveil the Presence of Structurally Diverse Oversulfated Contaminants"

_molecules, 2019, doi:10.3390/molecules24162988_

Round 1

Reviewer 1 Report

Authors showed results of characterization of crude and purified heparin batches contaminated with OSCS. In spite the number of papers already published on this matter, some of results obtained are original and interesting to better understand what happened in 2008.

However, some results appear to replicate data already described in other publications. Authors should cite previous studies, making comments on possible observed differences. For instance, the inhibition of heparinases by OSCS and other sulfated GAGs was shown and discussed by Aich et al in 2011 (Anal Chem 83, 7815). This paper should be cited and discussed.

The chemical shifts of proton signals of OSCS are strongly affected by the pH of solution (Nat Biotech 2010, 28, 207). Authors should report the pH of solution and based on previous published data assign all peaks of the OSCS spectrum of Fig. 5

According the COSY spectrum of figure 7, the proton signal at about 4.50 ppm highlighted with red asterisk should correspond to U2 of OSCS, while that at 4.55ppm to the  non-sulfation at 3-O position. Asterisk of figure 6 should be moved to the right signal. A more extended NMR study (simple TOCSY, HSQC and HSQC-TOCSY experiments) would enable a clear and complete assignment of these not fully sulfated ChS samples.

Too many experimental details are often reported in the text (lines 265-272, lines 386-397). Authors should move them from the text to the experimental section of figure captions.

Minor comments:

Line 48 The inhibitory effect o  is…..        remove o

Lines 308 and  316 ……1H NMR spectra show signals at the 4.55ppm region ….    4.0-4.55 regions

Author Response

We would like to thank the reviewer for the comments and suggestions that certainly added to the manuscript in its present format. Below are the answers to specific points raised by the reviewer.

However, some results appear to replicate data already described in other publications. Authors should cite previous studies, making comments on possible observed differences. For instance, the inhibition of heparinases by OSCS and other sulfated GAGs was shown and discussed by Aich et al in 2011 (Anal Chem 83, 7815). This paper should be cited and discussed.

New references were added to the manuscript. Nevertheless, it is relevant to emphasize that this is the first paper clearly showing that the contamination started when purifying heparin from porcine mucosa. Also, the results shown here demonstrate that the OSCS can have different degrees of sulfation. Regarding the inhibition of heparinase by OSCS the paper suggested in now cited and discussed regarding the similarities and differences in the results. Our assays differently from the work of Aich et al has been performed using both raw and final product heparins containing the OSCS. Also, we would like to elucidate the reviewer that the paper we are now citing has neglected all the papers published by our group. We were the ones that purified all the enzymes having also described their specificity.  

The chemical shifts of proton signals of OSCS are strongly affected by the pH of solution (Nat Biotech 2010, 28, 207). Authors should report the pH of solution and based on previous published data assign all peaks of the OSCS spectrum of Fig. 5

The spectra were recorded in unbuffered D2O. It is that in known that H5 of the GlcA shifts from around 4.1 ppm to about 4.2, which indeed is not our case. Even though we used unbuffered D2O, the signals obtained agree with those of OSCS close to neutral pH. We did conduct most of the more extended NMR analysis (TOCSY, HSQC and HSQC-TOCSY) and no significant differences were found. The major difference was found in the COSY spectrum were no overlapping was detected, hence we decided to present just that spectrum.

As requested, all the peaks of OSCS spectrum have now been assigned in Figure 5.

According the COSY spectrum of figure 7, the proton signal at about 4.50 ppm highlighted with red asterisk should correspond to U2 of OSCS, while that at 4.55ppm to the non-sulfation at 3-O position. Asterisk of figure 6 should be moved to the right signal. A more extended NMR study (simple TOCSY, HSQC and HSQC-TOCSY experiments) would enable a clear and complete assignment of these not fully sulfated ChS samples.

We thank the reviewer for this observation. The asterisks in Figure 6 indeed highlights the differences between the isolated contaminant and standard OSCS. They have now been placed in the correct position which is in accordance to the data shown in Figure 7, where a close-up region in the COSY spectrum is displayed. Again, in the case of the isolated contaminant, these signal point to the presence of CS samples in which complete sulfation was not achieved. We have included a new Figure 6 to the manuscript in its present format.

Too many experimental details are often reported in the text (lines 265-272, lines 386-397). Authors should move them from the text to the experimental section of figure captions.

Again, we thank the reviewer for the comments. The experimental details reported were part of the Figure legend.

Minor comments:

Line 48 The inhibitory effect o  is…..        remove o

Removed as indicated.

Lines 308 and  316 ……1H NMR spectra show signals at the 4.55ppm region ….    4.0-4.55 regions

Once more, we thank the reviewer for the suggestion, and it is now included in the present format of the manuscript.

Reviewer 2 Report

The manuscript presents interesting results. I have a request, that the authors present the anticoagulant activity of the different samples. It may be in vitro tests. I would like the authors to make it clear that the presence of the contans really did increase the anticoagulant activity of the sample. And if there is a correlation between the amount of contaminant and the anticoagulant activity of the samples.

Author Response

The manuscript presents interesting results. I have a request, that the authors present the anticoagulant activity of the different samples. It may be in vitro tests. I would like the authors to make it clear that the presence of the contans really did increase the anticoagulant activity of the sample. And if there is a correlation between the amount of contaminant and the anticoagulant activity of the samples.

First of all, we would like to thank the reviewer for the careful analyses and comments made to our work. The reviewer kindly asked us to include data on the anticoagulant activity of the different samples. Indeed, APTT assay shows that both crude heparin samples contaminate with OSCS or the final heparin product also contaminated with OSCS display lower anticoagulant activity when compared to pure heparin samples. We have now included these results as supplementary material. Accordingly, the sections of materials and methods, results and discussion were also modified to fit these new data. Similar results were also found by other authors and the papers are cited in the manuscript in its new format.

Please disregard the uploaded file. We were unable to delete this file.

Round 2

Reviewer 1 Report

Authors considered the reviewer's comment. The manuscript can be accepted for publication.